# Chyme Reinfusion in Intestinal Failure Related to Temporary Double Enterostomies and Enteroatmospheric Fistulas

**DOI:** 10.3390/nu12051376

**Published:** 2020-05-11

**Authors:** Denis Picot, Sabrina Layec, Eloi Seynhaeve, Laurence Dussaulx, Florence Trivin, Marie Carsin-Mahe

**Affiliations:** Réadaptation Digestive et Nutritionnelle, Clinique Saint Yves 4 rue Adolphe Leray CS54435, 35044 RENNES CEDEX, France; layec@clinique-styves.fr (S.L.); seynhaeve@clinique-styves.fr (E.S.); dussaulx@clinique-styves.fr (L.D.); trivin@clinique-styves.fr (F.T.); carsin@clinique-styves.fr (M.C.-M.)

**Keywords:** Intestinal failure, distal nutrition, parenteral nutrition, enterostomy, intestinal fistula

## Abstract

Some temporary double enterostomies (DES) or entero-atmospheric fistulas (EAF) have high output and are responsible for Type 2 intestinal failure. Intravenous supplementations (IVS) for parenteral nutrition and hydration compensate for intestinal losses. Chyme reinfusion (CR) artificially restores continuity pending surgical closure. CR treats intestinal failure and is recommended by European Society for Clinical Nutrition and Metabolism (ESPEN) and American Society for Parenteral and Enteral Nutrition (ASPEN) when possible. The objective of this study was to show changes in nutritional status, intestinal function, liver tests, IVS needs during CR, and the feasibility of continuing it at home. A retrospective study of 306 admitted patients treated with CR from 2000 to 2018 was conducted. CR was permanent such that a peristaltic pump sucked the upstream chyme and reinfused it immediately in a tube inserted into the downstream intestine. Weight, plasma albumin, daily volumes of intestinal and fecal losses, intestinal nitrogen, and lipid absorption coefficients, plasma citrulline, liver tests, and calculated indices were compared before and during CR in patients who had both measurements. The patients included 185 males and 121 females and were 63 ± 15 years old. There were 37 (12%), 269 (88%) patients with EAF and DES, respectively. The proximal small bowel length from the duodeno-jejunal angle was 108 ± 67 cm (*n* = 232), and the length of distal small intestine was 117 ± 72 cm (*n* = 253). The median CR start was 5 d (quartile 25–75%, 2–10) after admission and continued for 64 d (45–95), including 81 patients at home for 47 d (28–74). Oral feeding was exclusive 171(56%), with enteral supplement 122 (42%), or with IVS 23 (7%). Before CR, 211 (69%) patients had IVS for nutrition (77%) or for hydration (23%). IVS were stopped in 188 (89%) 2 d (0–7) after the beginning of CR and continued in 23 (11%) with lower volumes. Nutritional status improved with respect to weight gain (+3.5 ± 8.4%) and albumin (+5.4 ± 5.8 g/L). Intestinal failure was cured in the majority of cases as evidenced by the decrease in intestinal losses by 2096 ± 959 mL/d, the increase in absorption of nitrogen 32 ± 20%, of lipids 43 ± 30%, and the improvement of citrulline 13.1 ± 8.1 µmol/L. The citrulline increase was correlated with the length of the distal intestine. The number of patients with at least one liver test >2N decreased from 84–40%. In cases of Type 2 intestinal failure related to DES or FAE with an accessible and functional distal small bowel segment, CR restored intestinal functions, reduced the need of IVS by 89% and helped improve nutritional status and liver tests. There were no vital complications or infectious diarrhea described to date. CR can become the first-line treatment for intestinal failure related to double enterostomy and high output fistulas.

## 1. Introduction

Chyme is the mixture of the digestive secretions (saliva, gastric juice, biliopancreatic, duodenal, and intestinal secretions) with food being transformed into absorbable nutrients by enzymatic digestion and under the effect of bile salts. At the duodeno-jejunal flexure, the daily flow of the chyme is 6 to 10 L/24 h. At the end of the small intestine, the flow is only about 1 L/24 h containing sodium (around 100 mmol/L), indigestible or non-absorbable foods, some of which will be transformed by the colonic microbiota into useful nutrients and ultimate waste, constituting the stool.

An enterostomy is the emergence (stoma) of the small intestine (enteron) through the wall of the abdomen. Any abdominal surgery procedure may require the surgeon to perform a temporary double enterostomy (DES), especially if there has been a resection of the small intestine, anastomosis, peritonitis, mesenteric ischemia, radiation enteritis, etc. Entero-atmospheric fistulas (EAF) are intestinal fistulas in which the small bowel is completely open in a wound of the abdominal wall, with exposed external orifices (Figure 1). Only the part of the intestine upstream of the DES or EAF retains digestive and absorptive functions. The downstream bowel is no longer solicited for digestion, absorption, and citrulline synthesis, and the secretions of enterokines by the ileocolic L-cell enterocytes are reduced and their regulatory functions are interrupted. Enterohepatic cycles are interrupted, and the malabsorption of bile salts causes their hepatic secretion to run away, aggravating diarrhea and contributing to intestinal failure-associated liver disease (IFALD). Little is known about the absorption of most drugs, which differs from patient to patient and from drug to drug, some of which have an enterohepatic cycle.

Short bowel syndrome (SBS) Type 1 is defined as a length of the small intestine less than 200 cm between the duodeno-jejunal angle and the proximal enterostomy [1]. Intractable high outputs of chyme can lead to intestinal failure (IF) defined by European Society for Clinical Nutrition and Metabolism (ESPEN) as “the reduction of gut function below the minimum necessary for the absorption of macronutrients and/or water and electrolytes, such that intravenous supplementation is required to maintain health and/or growth” [2]. The ESPEN functional classification describes a Type 2 IF (IF2) as a “prolonged acute condition, often in metabolically unstable patients, requiring complex multidisciplinary care and intravenous supplementation over periods of weeks or months” [2]. According to required volumes in the ESPEN clinical classification, intravenous supplementations (IVS) are FE type if they provide only fluids and electrolytes for hydration and PN type if they are parenteral nutrition [3].

DES and EAF can result in a Type 2 IF per Type 1 SBS. It is a rare and temporary form of a rare disease; its incidence is poorly known and appears to be less than 10/million inhabitants/year. Saunders estimated the costs at 5000–10,000 €/month in 2012 [4]. Restoration of continuity requires a new surgical operation three to six months after the initial surgery and may exceed one year. The jejunum is the part of the body most physiologically required to absorb nutrients and its ability to adapt is limited within these delays.

IVS is the reference treatment in hospital and at home while awaiting surgical anatomical closure. They remain dangerous [5] and are not available in many countries or at the risk of a “financial catastrophe” for patients and their families [6].

Chyme reinfusion (CR), developed by Dr Etienne Levy in the 1970s, establishes an extracorporeal circulation of the chyme between the collection pouch and the downstream small intestine. This corrects intestinal failure, restores enterohepatic cycles, and stimulates L-cell enterocytes in proportion to the additional function that the downstream intestine is capable of performing. More than 90% of patients can stop IVS while improving nutritional status and IFALD. The bowel function achieved prefigures the outcome of surgical restoration of the anatomical continuity of the bowel.

CR is one of several “distal nutrition” techniques, all of which can restore digestive function to the downstream intestine and reduce the need for IVS. ASPEN [7] and ESPEN [8,9,10] have been recommending them for some years. 

We report the prospective observational data of entero–enteral CR performed from January 2000 to December 2018 in our center. These data were published as posters at the 16th Congress of the Intestinal Rehabilitation and Transplant Association (CIRTA 2019) in Paris [11]. The current publication complements a previous publication in 2017 [12]. Our results shed light on certain points of the pathophysiology of DES and confirm the effectiveness and safety of the technique. The aim is to encourage caregivers to be trained in this technique and medical manufacturers to offer devices that are effective and easy to use.

## 2. Materials and Methods

We report a prospective observational cohort of consecutive patients with IF2 secondary to temporary DES or EAF, specifically referred for CR to the unit of nutritional and digestive rehabilitation from January 2000 to December 2018. Our unit is a tertiary center and is a competent center for the management of rare digestive diseases in adults (MaRDi) and a certified home parenteral nutrition center for adults. 

All patients admitted for DES or high-flow small bowel EAF were candidates for CR if several of the following conditions were met: the average flow rate of the bowel was at least 1200 ml/24 h; the upstream and downstream holes had to be visible; the downstream bowel had to be accessible and to have at least 15 cm of healthy small bowel; the upper and efferent intestinal tract had to be free of fistulas, active inflammatory diseases, stenosis, obstacles, or active cancer processes; a feeding tube had to be placed in the intestine downstream up to 15–25 cm from the enterostomy orifice; in cases of well-drained abdominal abscess, CR was contraindicated if the abscess was the result of a fistula or if it caused transit disorders or intestinal paresis; and the surgical closure of the intestine had to be planned. After explanation of the technique, the patient gave full agreement to perform the CR and accept the dietary constraints (food must be pureed).

### 2.1. Data Collection

The clinical and biological data were prospectively recorded. The dedicated Access database has been registered at the French Committee for Computing and Freedom (CNIL; N° 1452427).

At admission, height was measured or, if necessary, estimated from the knee height according to Chumlea equation. We measured weight twice a week and enterostomy or fistula outputs and urine on a daily basis. During the CR, feces were weighed only to check for diarrhea i.e., number was ≥3/day or easy to collect (ileostomy or colostomy). In all patients, blood samples were collected prior to the onset of the CR. Subsequent monitoring, during the CR, was prescribed based on the clinical course. Plasma values for alanine aminotransferase ALT, aspartate aminotransferase AST, alkaline phosphatase and γ-glutamyltransferase γGT were reported at the upper normal (N) values of each and were considered pathological if >2N. The following indices were calculated: weight loss (%) at admission (100 *(usual weight – actual weight) / usual weight)); body mass index, (BMI; weight (kg)/height (m)²) and Nutritional Risk Index (NRI; 1.519*plasma albumin+41.77* actual weight/usual weight). An NRI > 97.5 was defined as no risk of malnutrition and NRI < 83.5 defined a risk of severe malnutrition.

The dietician recorded oral ingesta twice a week using Nutriciel® (SCJ informatique, 27 rue Alfred Kastler, 76130 Mont Saint Aignan, France), a computer application that compares nutrient intake (energy, protein, carbohydrate, fat) from meal orders with actual intakes based on visual estimates of portions consumed (nothing, 25%, 33%, 50%, 66%, 75%, all). The sum of oral intakes and enteral supplements received per day was expressed in g protide/kg and kcal/kg actual body weight.

When it was possible and necessary, intestinal nitrogen losses and steatorrhea were measured over three consecutive days and were compared to contemporary intakes. Nitrogen concentration was measured according to Kjeldahl’s method and the steatorrhea by Van de Kamer’s methods. Contemporary oral and enteral protein and fat intakes were determined. Intestinal nitrogen and fat absorption coefficients were the percent of protein and fat intake not recovered in intestinal losses. They were calculated as: nitrogen absorption = (1–(intestinal nitrogen losses (g/day)/ protein intake (g/day)/6.25)*100; and lipids absorption = (1–(steatorrhea (g/day)/ fat intake (g/day))*100. Normal values are greater than 90%, but values above 85% were considered clinically sufficient.

Since 2008, the plasma concentration of citrulline at 8:00 a.m. was determined before CR onset and during CR, using reverse-phase HPLC. 

The length of the proximal intestinal segment, from duodeno-jejunal flexure to the afferent stoma was determined by the surgical report or, exceptionally as measured on X-ray documents obtained with opacification. The distal small bowel length was determined by the surgical report or X-ray opacification from the distal stoma to the end of the distal small bowel, i.e., the ileo-caecal valvula, an ileo-colic or ileo-anal anastomosis, or a terminal ileostomy. By adding the proximal and distal lengths, when both were known, the theoretical length of the total useful small intestine was obtained.

### 2.2. Chyme Reinfusion

CR consisted in an extra-corporal circulation of chyme in a closed system. We used two sorts of pumps. Enteromate 2 (Société Labodial, 30 rue des Dames 78340 Les Clayes-Sous-Bois, France) is an automaton that constantly aspirates the chyme and injects it into the efferent small intestine by adapting the flow rate to the quantity aspirated so that the pouch remains empty. Enteromate Mobile (Société Labodial, marketed from 2010) is a battery-powered portable pump, whose flow rate is regulated by the patient or caregivers using a rheostat. A Levin type F14-16 polyurethane tube, without balloon, was placed in the first 15–20 cm of the downstream small intestine. An X-ray opacification checked the tube position, looked for an abnormality contraindicating CR, and measured the length of the distal bowel (Figure 2). In the tube, we performed enteroclysis with 1 L of oral rehydration solution for 24–48 h, with added laxative if stools have been seen in colon. The CR started with an Enteromate 2. After a few days, CR was continued with the Enteromate Mobile if the patient was able to use it. With both devices, the circuit was closed, odorless, and did not require manipulation of the chyme. Enteromates are no longer manufactured.

### 2.3. Nutrition

Unless impossible, patients were given oral nutrition. Food had to be prepared as purees to avoid clogging the tubing. If necessary, supplemental enteral nutrition was delivered through a naso-gastric feeding tube, gastrostomy, jejunostomy (when in place), or “en Y” enteroclysis in the reinfusion tube in the downstream small bowel. 

Intravenous supplementations (IVS) have been classified into FE and PN according to the ESPEN clinical classification. Patients who received IVS for more than two weeks after CR onset and who were not weaned prior to the last week prior to discharge, were considered “dependent” on IVS during the CR. 

### 2.4. Statistical Analyses

Statistical analyses were performed with Excel spreadsheet program and Statistical Tools for High-Throughput Data Analysis (http://www.sthda.com/). The durations were expressed as median (quartile 25–75%) and the other data as mean ± standard deviation (SD). The normality of data distribution was analyzed by the Smirnoff–Kolmogorov test. The means of paired data before and during CR were compared by paired Student’s *t*-test. Comparisons of means between groups were made using the unpaired Student’s *t*-tests. Differences of distributions of the NRI groups were studied by Chi-squared test. *p*-values less than 0.05 were considered as statistically significant. Pearson linear correlation were investigated between citrulline and small bowel lengths, and between citrulline prior and during CR. *p*-values equal to or less than 0.05 were considered statistically significant.

Ethical statement. This work was conducted in accordance with the declaration of Helsinki. Our study was a non-interventional, retrospective observation of clinical and biological data on the use of a validated medical treatment since the 1970s. The study has in no way altered or guided treatment. The data were anonymized and collected from 01/2000 to 12/2018 on a specific database declared to the CNIL (Commission Informatique et Libertés) under the number 1452427. Patients have been informed of the existence of this database and the nature of the data stored in memory and have given their consent. All the accreditation visits carried out in our unit since 2000 by the French High Authority for Health (HAS) in 2002, 2007, 2011, 2015, and 2019 have validated the compliance of our provisions with regulations.

## 3. Results

From January 2000 to December 2018, 306 consecutive patients had CR. Some patients had to discontinue CR due to ileal stenosis, anastomotic stenosis, downstream segment fistula, active cancer, mesenteric angina, ischemic colitis, inflammatory enteritis, or disabling anal incontinence. Demographics and main etiologies of small bowel surgery are summarized in Table 1.

Missing data was less than 1% for weight and height (thus BMI), as well as oral, enteral, and parenteral nutrient intakes. Additionally, missing data was less than 2% for initial albumin and liver test values, 14% for usual weight, 8% for daily DES or EAF output, and 24% for citrulline prior CR values (available since 2008). The pairs of values before and during CR were: albumin values 90%, liver tests 79%, stool weight 84%, citrulline 41%, nitrogen absorption coefficients 21%, lipid absorption coefficients 14%, and NRI 90%. 

Consequences of surgical procedures on the small bowel anatomy are shown Table 2. A wide variety of abdominal pathologies required 269 (88%) temporary DES to be performed, especially after perforation, bowel resection, or to protect an anastomosis in the downstream bowel. These are most often double enterostomies located in the same parietal orifice (87%). All EAF were postoperative, mainly (*n* = 30, 81%) due to peritonitis or mechanical occlusion. In both cases, volvulus in wall hernias or perforations on parietal prosthesis material were frequent (*n* = 12/30). One patient, who had a post-operative recurrence of EAF, was treated with CR for both periods. In addition to small bowel surgery, some digestive organs have been resected during the same or previous surgery. In 106 (35%) patients, resection of the ileocolon (29), transverse or left colon (39), whole colon (19), or rectum (19), of which with a terminal ileostomy (29), was performed. Twenty-one (7%) underwent Lewis-Santy oesogastrectomy (3), Finsterer or total gastrectomy and Roux-en-Y reconstruction for cancer (15), or gastric bypass (3).

The length of the proximal segment of small bowel was known or estimated in 232 (76%) patients. It was ≤100 cm in 127 (55%) and ≤50 cm in 51 (22%) patients. The lengths of the distal segment of small bowel were known or estimated in 253 (77%) patients and both lengths in 197 (64%). Theoretical total length was less than 200 cm in 56 (28%), distributed in SBS Type 1, 2, and 3, *n* = 8, 7, and 41, respectively. Six patients had total length ≤100 cm and none less than 50 cm. Eight patients had intermediate segment longer than 120 cm ending in a second DES or EAF. They were treated by a double CR.

During CR, intestinal function improved dramatically (Figure 3). The intestinal losses were reduced by 86% (*p* < 0.000001). Intestinal nitrogen and lipids absorption improved by 73% and 91%, respectively, and steatorrhea decreased by 63%. Citrulline increases in 91 patients who had the two measures. The median time between initiation of CR and plasma citrulline measurement was 27 days (16–43). The number of patients with citrulline with a normal level >25 µmol/L increased from 1667%. Citrulline prior CR was moderately correlated with the proximal bowel length (*r* = 0.214, *p* = 0.018, *n* = 122 pairs). Citrulline during CR was correlated with the distal bowel length only in patients with <200 cm (*r* = 0.494, *p* = 0.0195, *n* = 22). The increase in citrulline was correlated with the distal length (*r* = 0.257, *p* = 0.034, *n* = 68), especially if it was <200 cm (*r* = 0.317, *p* = 0.010, *n* = 64), but it was not correlated with the reduction of intestinal losses nor improvement of the nitrogen and lipid absorption coefficients. Citrulline during CR was strongly correlated with citrulline prior CR (0.484, *p* < 0.0001, 91 pairs). 

Nutrition. During CR, the oral and enteral nutritional intakes were known for 292 patients: 30.0 ± 12.3 kcal/kg/d, 1.26 ± 0.52 g/kg/d proteins, and 8 (3%) had no oral intakes and received an exclusive enteral nutritional (EN). The oral intakes were exclusive in 171 (59 %) patients (35.0 ± 10.3 kcal/kg/d, 1.50 ± 0.42 g/kg/d proteins). In 122 (42%) an enteral supplementation was provided for nutrition (*n* = 114) or hydration alone (*n* = 8) and their total intakes were 34.9 ± 9.3 kcal/kg/d, 1.58 ± 0.41 g/kg/d protein. Enteral supplementation was provided via gastrostomy (*n* = 2–6%), jejunostomy (*n* = 1–2%), nasogastric tube (*n* = 5–9%), and/or via “en Y” enteroclysis in the downstream segment tube (*n* = 30–87%). 

At admission, 211 (69%) patients were receiving IVS, divided into 162 PN and 49 FE. Mean ± standard deviation (SD) energy and protein PN were 24.1 ± 7.9 kcal/kg/day and 0.97 ± 0.29 g/kg/day, respectively, in a mean volume of 2359 ± 941 mL/day. FE provided 178 ± 95 mmol/day of Na^+^ in 1420 ± 884 ml/day. IVS was stopped in 188 (89%) patients, within a median (Q 25–75%) of two (0–7) days after CR had begun. During CR, three patients needed FE for hydration alone and 20 for a PN supplement (Figure 4). The median duration without PN was 64 (45–92) days.

Patients (23/211, 11%) who required IVS during CR had a shorter small bowel total length and, during CR, a lower citrulline, less diarrhea, and a more severe malabsorption (Table 3). There were no significant differences in percent weight loss, BMI, albumin, NRI, or liver tests.

At admission, 96 patients had no IVS. Compared to patients receiving IVS, weight loss was greater (14.7 ± 10.0% vs. 11.2 ± 8.9% *p* < 0.01), DES or FAE flow rates were slightly lower (2210 ± 745 mL/24h vs. 2524 ± 2210 mL/24h, *p* = 0.019), and citrulline before CR was significantly higher (23.5 ± 11.9 µmol/L vs. 18.9 ± 9.5, *p* = 0.010). Moreover, proximal length (131 ± 81 cm, vs. 97 ± 56, *p* = 0.0003) and total length (255 ± 85 vs. 220 ± 84, *p* = 0.007) were longer. There were no other significant differences (BMI, oral ingesta, enteral intakes, albumin, liver tests, nitrogen intestinal absorption, and steatorrhea). We considered them to be clinically in a state of IF2.

With respect to nutritional status, the majority of patients had one or several malnutrition criteria when they were admitted to our unit (Table 4). Weight increased by 3.5 + 8.4% (*p* < 0.0001) during CR for all patients. Weight gain was significant in patients with an initial BMI <20 (9.87% ± 11, *p* < 0.00001, *n* = 79) and for BMI 20–25 (2.42% ± 6; *p* < 0.00001, *n* = 135), but was not significant for BMI 25–30 (0.82% ± 5, *p* = 0.32, *n* = 60). Additionally, patients with a BMI >30 lost weight (0.64% ± 7, *p* < 0.00005, *n* = 30).

C-Reactive Protein was higher at the beginning (45.5 ± 44.5 mg/L) than at the end (11.6 ± 11.7 mg/L, *p* < 0.00025). Nutritional status improved, with a 65% reduction of the number of patients at risk of severe undernutrition (NRI < 83.5).

Plasma liver tests were paired in 237 patients (Figure 5). The number of patients with one or more test >2N decreased significantly from 29–13%, 11–4%, 48–13% and 81–40% for AST, ALT, Alkalin phosphatase, and γGT , respectively. The number of patients with one or several liver tests abnormalities decreased from 83–40% (*p* < 0.0001). There was no significant difference in liver tests between patients with or without IVS at admission, nor in percentage of patients with at least one value >2N (79% vs. 86%).

Median delay (quartile 25–75%) between the initial surgery and admission in our center was 31 (19–7) days. CR began five (2–10) days after admission and lasted 64 (45–94) days, max 272 days. The end of CR corresponds to the date of the closure surgery. The median time (days) between the two surgical operations was 104 (85–151) and was longer for EAF than for DES patients (153 (109–206) vs. 101 (84–136), *p* < 0.01). The median hospital stay after intestinal closure was nine days (6–13), known for 186 patients. 

CR was pursued at home by 81 (56 males and 25 females) patients since 2007, for a median duration of 47 days (28–77) and max of 187 (Figure 6). They all had DES, none of the patients with EAF benefited from it. Patients with home CR were younger (51.8 y ± 15.6 vs. 66.8 ± 13.2, *p* < 0.0001) and at baseline, they had a higher albumin (31.3 g/L ± 6.4 vs. 28.3 ± 6.0, *p* = 0.0012) and their upstream small bowel length was longer (129 cm ± 65 vs. 101 ± 66, *p* = 0.0043). No patient had to stop CR. 

## 4. Discussion

CR as a treatment of temporary IF2 associated with high outputs DES or EAF is very rarely reported in the literature [13]. To our knowledge, our cohort is the largest on the topic. It is also the most documented in terms of the effects of CR on intestinal function, the oral ingesta, the needs of enteral and parenteral supports, the possibility of weaning from IVS, the improvements of nutritional status, and liver function.

There was little missing data in the results of weight, albumin, and liver tests. There is less data on citrulline, as it was only available from 2008. In patients who did not have diarrhea during the CR, citrulline, stool weight, nitrogen, and lipid absorption coefficients were not measured. Delays between the onset of CR and measurements were variable, which may lower the significance of statistical relationships, particularly for citrulline.

Patients with DES or EAF have two enterocyte masses, one in function and the other at rest. The sum of the anatomical lengths of the upstream and downstream segments gives an indication of the length of the functional small intestine during CR and after the surgical closure. When that total length remained less than 200 cm, CR and closure surgery transformed a short bowel syndrome (SBS) Type 1 into a longer and less severe SBS Type 1, 2, or 3 in 56 cases.

Citrulline is the biomarker of functionally active enterocyte mass. It correlated well with the lengths of corresponding segments of small bowel, citrulline prior CR with proximal length, and citrulline during CR with total length. Interestingly, the increase in citrulline correlated with the increase of the small bowel length i.e., the length of the distal intestinal segment. This result was expected but had not yet been shown, as far as we know. Correlations of citrulline are best with lengths <200 cm. The correlation of citrulline with the length of the small bowel was low in SBS Type 1 [14]. The correlation between length and citrulline would not be linear and this would explain why they are better in the lower length values [15]. It is likely that citrulline is correlated with additional length of the small intestine until bowel function and plasma citrulline are normal. Beyond that point, an additional length of small intestine no longer increases citrulline, possibly because all of the dietary glutamine has been absorbed and utilized. The citrulline generation test by an oral load of glutamine proposed by Peters [16] would be a clinically relevant test. We found a strong correlation between citrulline prior and during CR. Jeppesen [14] reports an increase and a strong correlation between baseline and week 24 citrulline, under teduglutide treatment. This improvement in the function of the upper intestine without increasing its length can be explained by several mechanisms. Levy has shown that CR reduces daily losses of the upstream enterostomy by 30% [17]. Restoring the enterohepatic cycle of bile salts stops their hypersecretion and their caustic and laxative effects that aggravate diarrhea. CR restores the post-prandial increase in Glucagon-Like Peptide-2 secretion by L-cell enterocytes, which is the major growth factor in the intestinal epithelium [18].

Patients were fed only by mouth in 59% of cases during CR and their caloric and protein ingesta were satisfactory. When oral intakes were insufficient, enteral supplementation was administered. “en Y” enteroclysis in the CR tube is easily performed. Nutrient intakes, sum of oral, and enteral, were also satisfactory.

The incidence of IF is underestimated in patients with enterostomy. The 96 patients who did not have an IVS on admission were at high risk of dehydration and undernutrition due to the volume of losses and intestinal malabsorption. Dehydration accidents occur in 16–20% of patients with enterostomies, are the cause of 40% of readmissions [19], and the relative risk of acute renal failure is 2.4 [20]. IVS weaning was achieved in 188/211 (89%) of cases. The remaining 23 patients required lower volumes of IVS, mainly PN (20/23). The total length of their small bowel was shorter, and they exhibited symptoms of intestinal failure or deficiency during CR. (Table 3). Spontaneous weaning from IVS is possible in some patients without CR but Type 1 SBS is the least suitable situation for adaptation of the small intestine in a short period of three to four months.

Nutritional status improved mainly for malnourished patients. This is not a specific effect of CR, as IVS also achieves this. Patients are fed by what they eat, in most cases. Several publications describe distal enteral nutrition by enteroclysis without CR [21,22]. The results are often sufficient to wean the majority of patients from IVS. Intestinal failure-associated liver disease (IFALD) is multifactorial and regresses during CR. The association with IVS is not confirmed as, at admission, neither the values nor the prevalence of liver test abnormalities differed statistically between patients who received IVS and those who did not. These abnormalities appear to be more a consequence of the interruption of the small intestine with IF2 than of the composition in parenteral nutrients. Reinfusion of intestinal secretions or bile alone, with distal feeding, and without oral feeding improve IFALD [23].The preliminary results of our prospective study RESCUE (Restored Enterohepatic Signaling: Chyme ReinfUsion ThErapy) [24] showed that plasma levels of fibroblast growth factor 19 (FGF19), secreted by the L-cells enterocytes, were very low prior to CR and increased sharply at the onset of CR, and at the same time, plasma markers of bile salt hypersecretion collapsed. CR restores enterohepatic signaling that inhibits the hypersecretion of bile salts and would thus contribute to the IFALD improvement.

The economic aspect is not negligible. Nagar estimates the cost of the handmade CR with a syringe at a few rupees per day [6]. The cost of CR was estimated at 34.8 €/day/patient in our unit. Regardless of their nature, IVS is much more expensive [25].

Chyme reinfusion can be done at home. Therapeutic education of patients and their relatives is rapid and is done without delaying discharge. A recipe booklet is provided so that they can continue to eat pureed food. We believe some home care providers would be interested in the home RC but the lack of reimbursement by health insurance does not allow them to do so. Currently, only very autonomous patients can have a home CR, with pumps lent by our institution, under close supervision by telephone and teleconsultation and sometimes with the voluntary help of home care providers.

CR is very rarely performed due to the lack of pumps and devices specifically designed for avoiding the handling of chyme and bad odors, which are considered repugnant by health care staff and patients. Care can be disgusting when bad methods are used. In the Du Toitab study [26], patient resistance to use distal nutrition was 8% and caregiver resistance was 75%. Pedagogy is needed.

CR is a safe technique if its contraindications are respected and if the protocol of care is well observed. There are no vital complications or infectious diarrhea described to date. Side effects are mainly abdominal colic, which is frequent if CR is performed by bolus or after cooling of the chyme, and diarrhea due to bile salts during the first days, which is relieved by cholestyramine. Bacterial translocation may occur during the first few hours of the hydration enteroclysis done prior to CR. It is prevented by adding a non-absorbable antibiotic to the enteroclysis solution when there is residual stool in the colon. Some CR symptoms or complications prefigure the outcome of the planned surgical closure. Residual intestinal failure or deficiency can be measured, and post-operative management can be planned. Anal incontinence can be identified, and subsequent rehabilitation can be initiated. Some complications can appear after the operation, mainly ileus, fistulas, and intractable incontinence, and can require a new operation, whereas stopping CR is almost always sufficient. In some cases, it can result in a change in the planned surgery. CR contributes to adaptation or prevents disadaptation of the downstream intestine and reduces the incidence of postoperative ileus associated with surgical closure of enterostomies [27]. In the future, CR may reduce the time required to restore continuity by showing that intestinal functions have been restored and are stable.

The FRY (efFiciency of Reinfusion of the chYme) study [28] is a prospective, randomized, controlled, multicenter study whose primary objective is to evaluate the effect of CR versus IVS treatment on the frequency of complications occurring from the day of inclusion to 30 days after intestinal closure surgery in patients who have undergone double temporary high flow enterostomy. This study requires that all investigators use the same device for continuous reinfusion. Interruption in the manufacture of Enteromates prevented the study from being started. Several teams are working on new chyme reinfusion devices [29,30].

Quality of life (QoL) was not measured by a QoL scale. Our impression is that if bowel function is normal, CR occurs “in the silence of the organs”. The technical constraint is the weight of the Enteromate Mobile carried in a bag (2 kg). The main constraint is the feeding of puree, which is felt as monotonous and unappetizing. At home, patients prepare their meals and know what they are eating. Fewer than 5% patients asked for CR to be stopped and IVS be restarted. 

Knowledge of the effects of CR comes exclusively from clinical practice. There is no animal model in which the distal part of the intestine is isolated and in which the effects of CR can be measured non-invasively. The influence of the chyme on the ileocolic absorption of nutrients, drugs, and bile salts on the production of enterokines, on the microbiota, and on IFALD remains to be explored. 

## 5. Conclusions

Temporary DES and EAF may result in a Type 2 intestinal failure. Parenteral supplementations compensate for nutrients, water, and electrolyte losses. By artificially restoring the continuity of the remaining intestine, CR restores all the functions it is capable of. Most often, patients are nourished by what they eat, and nutritional status and liver test abnormalities improve, and complete cessation or drastic reduction in IVS requirements lowers the risk of venous central line complications. CR is recommended by ESPEN and ASPEN whenever possible. It is now time to consider CR as the first-line treatment for IF2 associated with DES and EAF. 

## Figures and Tables

**Figure 1 nutrients-12-01376-f001:**
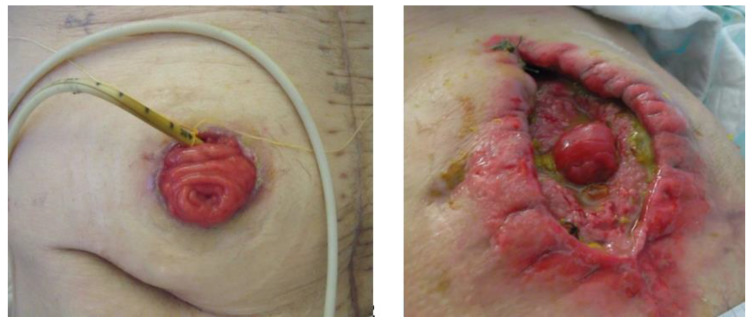
Double enterostomy (left) with a tube placed in the downstream intestine to reinfuse the chyme. Entero-atmospheric fistula (right) where the small intestine is poured out of the abdomen, located in a wound in the wall and behaves like a double enterostomy.

**Figure 2 nutrients-12-01376-f002:**
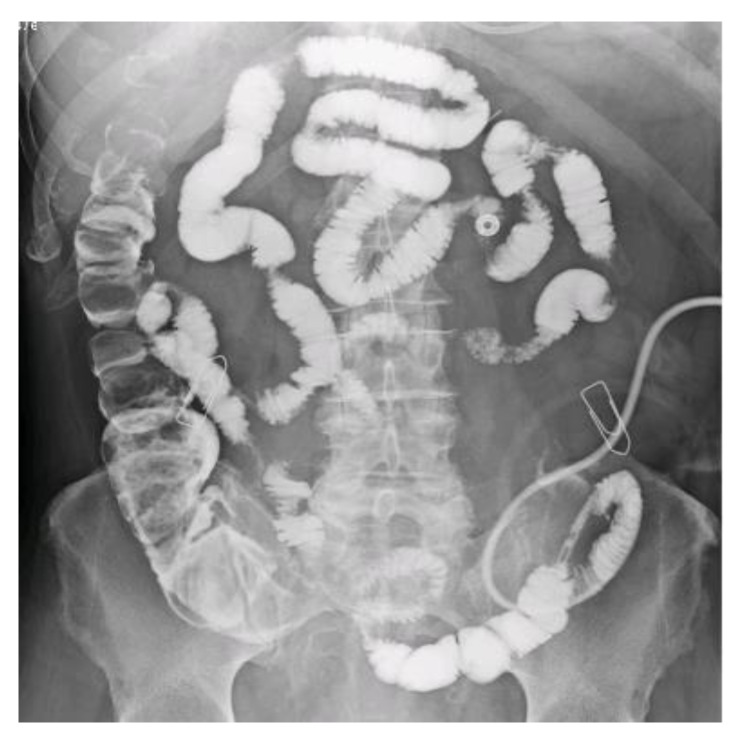
Example of distal bowel opacified by the chyme reinfusion tube inserted 25 cm from the downstream enterostomy (designated by a paper clip). The length was greater than 250 cm. On the right flank, the paper clip refers to the terminal transverse colostomy.

**Figure 3 nutrients-12-01376-f003:**
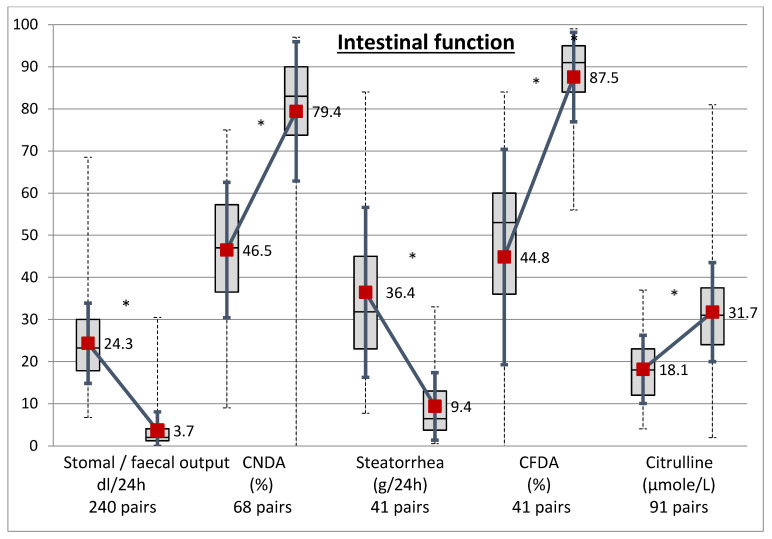
Comparison of intestinal function before (left boxes) and during (right boxes) chyme reinfusion. CNDA, coefficients of nitrogen digestive absorption; CFDA, coefficient of fat digestive absorption. * *p* < 0.000001.

**Figure 4 nutrients-12-01376-f004:**
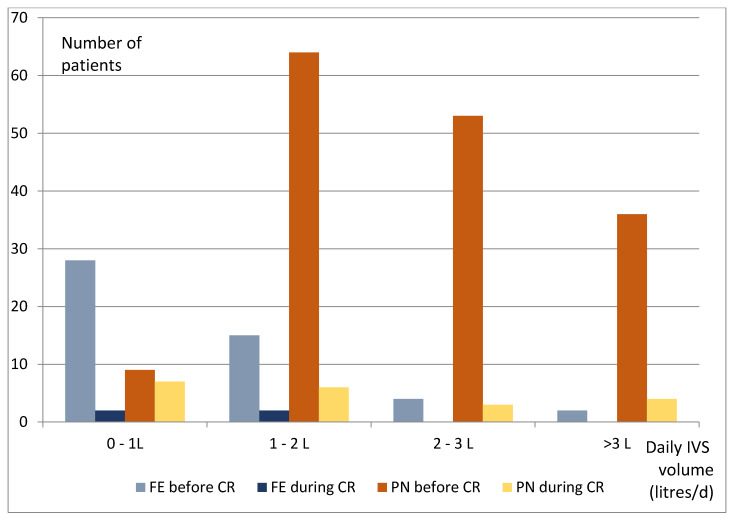
Daily volume of the intravenous supplementations (IVS) before and during chyme reinfusion (*n* = 211 patients), in liters (L). CR, chyme reinfusion; FE, fluids and electrolytes; PN, parenteral nutrition.

**Figure 5 nutrients-12-01376-f005:**
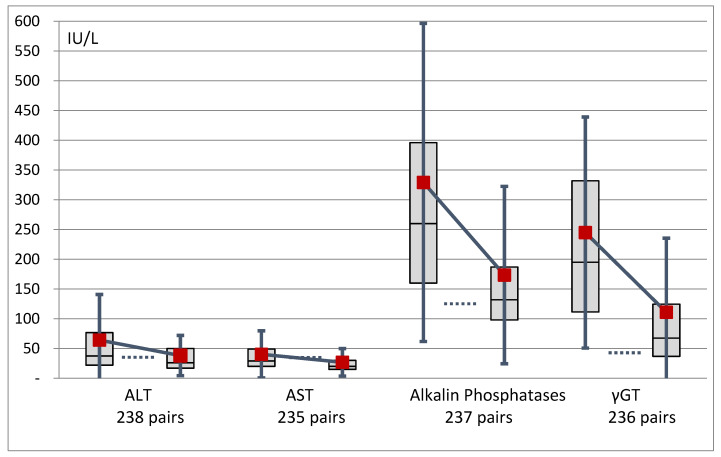
Evolution of liver tests expressed as their international units. The dashed horizontal bars represent the upper values of the normal for each data pair. *p* < 0.0001 for all. ALT: alanine aminotransferase; AST: aspartate aminotransferase; γGT: γ-glutamyltransferase.

**Figure 6 nutrients-12-01376-f006:**
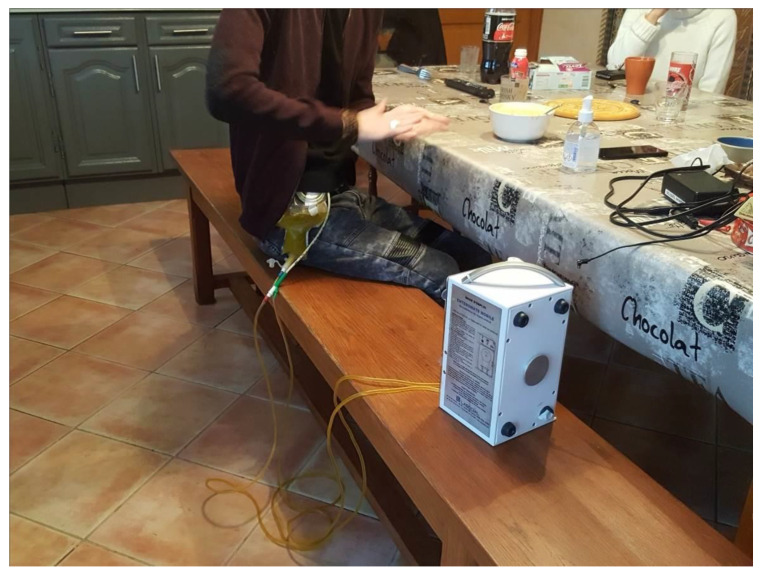
Male, 21 years old, hospitalization 27 days, CR at home 153 days.

**Table 1 nutrients-12-01376-t001:** Patients’ demographics and pathology (*n* = 306).

	Double Enterostomy	Entero-Atmospheric Fistula
Male/female ^#^	159 (59)/110 (41)	26 (70)/11 (30)
Age (years) ^§ #^	62.5 ± 15.8; 17 (93)	65.0 ± 12.3; 30 (98)
Pathology^#^		
Cancer	39 (14)	6 (16)
Radiation enteritis	10 (4)	
Mechanical occlusion	75 (28)	12 (32)
Ischemia	59 (22)	
Peritonitis	64 (24)	18 (49)
Inflammatory bowel diseases	16 (6)	
Trauma	6 (2)	1 (3)

^#^*n* (%). ^§^ mean ± standard deviation (SD; range).

**Table 2 nutrients-12-01376-t002:** Consequences of surgical procedures on the small bowel anatomy.

	Double Enterostomy *n* = 269 (88%)	Entero-Atmospheric Fistula *n* = 37 (12%)
Loop stoma ^#^	57 (21)	29 (78)
Double end-loop stoma ^#^	177 (66)	
Separated double end stoma ^#^	35 (13)	8 (22)
SB resection ≥30 cm ^#^ SB resection 30–99 cm ^#^; ≥100 cm ^#^	136 (51) 88 (47); 48 (25)	12 (32) 4 (33); 4 (33)
Afferent SB length (cm) ^§ #^ Afferent SB length ≤ 150 cm ^#^	112 ± 67; 211 (78) 172 (82)	66 ± 49; 21 (57) 19 (90)
Efferent SB length (cm) ^§ #^	115 ± 70; 225 (84)	133 ± 80; 28 (76)
Total theorical SB length (cm) ^§,#,^*	232 ± 87; 178 (66)	225 ± 74; 21 (57)
Total theorical SB length ≤ 150 cm ^#^	32 (18)	3 (15)
Downstream SB anatomy ^#^		
Ileo-colon	205 (75)	23 (62)
Terminal ileostomy	22 (8)	7 (19)
Ileo-rectal anastomosis	8 (3)	1 (3)
Terminal colostomy	27 (10)	5 (14)
Double chyme reinfusion	7 (3)	1 (3)

^§^ mean ± SD. ^#^ number of patients (%). * Total SB length = afferent + efferent SB lengths if both lengths are known. SB, small bowel.

**Table 3 nutrients-12-01376-t003:** Patients who required intravenous supplementations (IVS) for more (IVS+) than 14 days or less (IVS–) after the start of CR.

	IVS + (*n* = 23/211)	IVS − (*n* = 188)	*p*
Total small bowel length (cm) ^§^	167 ± 60 (17)	237 ± 85 (179)	0.0010
Plasma citrulline during CR (µmol/L) ^§^	23.2 ± 154.7 (9)	32.5 ± 11.4 (95)	0.025
Fecal output during CR (g/24) ^§^	1058 ± 847 (20)	236 ± 314 (236)	<0.0001
CNDA during CR (%) ^§^	68.7 ± 17.9 (13	82.8 ± 103 (132)	<0.0001
CFDA during CR (%) ^§^	71.8 ± 17.7 (8)	85.7 ±11.6 (99)	0.0019

^§^ Mean ± SD (*n*). IVS, intravenous supplementations; CR, chyme reinfusion; CNDA, coefficient of nitrogen digestive absorption; CFDA, coefficient of fat digestive absorption.

**Table 4 nutrients-12-01376-t004:** Evolution of nutritional status between before and during chyme reinfusion (CR).

	Before CR	During CR	Number of Pairs
Weight loss > 10% *n* (%)	152 (58)	110 (42)	263
Body mass index ^§^	23.4 ± 5.4	23.9 ± 4.8	305
Plasma albumin (g/L) ^§^	28.8 ± 6.2	34.1 ± 4.9	274
Plasma albumin < 30 g/L	162 (59)	58 (21)	274
NRI ^§^	78.8 ± 10.1	88.1 ± 8.8	237
NRI < 83.5	164 (69)	62 (26)	
83.5 ≤ NRI ≤ 97.5	68 (29)	148 (62)	
NRI > 97.5	5 (2)	27 (11)	

^§^ mean ± standard deviation. NRI, nutritional risk index. *p* value < 0.0001 for all.

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
