# Peer review of "Chyme Reinfusion in Intestinal Failure Related to Temporary Double Enterostomies and Enteroatmospheric Fistulas"

_nutrients, 2020, doi:10.3390/nu12051376_

Round 1

Reviewer 1 Report

The authors present a very nice manuscript describing their experience with chyme re-infusion. Is a very well presented paper and I found a few areas that can be improved, some of them related with language and some content

An overarching comment is that the large number to abbreviations that are difficult to follow so may want to limit them for improvement if the flow of the paper.

Second the impact of the lack of Enteromates pumps needs to be fully explored, personally I have used the technique many times but never without significant resistance from home health provides, this does not seem to be a big factor for the authors and may reflect on this specialized pumps

Third please see below:

Line 26 should be 185M not H

Line 63 are L cell enterocytes or enteroneuroendocrine cells

Line 78 the authors have quotations, but I am unsure what is the reference for those quotations.

Line 82 same

Line 99 same as in line 63

For table 2 consider changing stomy for stoma

Line 239 consider using resected instead of amputated

Line 242 what procedure is specifically described as oesogastrectomy,  is a total gastrectomy with distal esophageal resection, a esophagogastrectomy (Ivor Lewis, three field or Tranhiatal type?)

Line 262 space is missing before the word Prior

Line 272 and line 297 consider changing Stomial for Stomal

Line 357 to 363 is difficult to read

Line 385 and thereafter consider change the word ingesta for intake although both are accurate

Author Response

The authors present a very nice manuscript describing their experience with chyme re-infusion. Is a very well presented paper and I found a few areas that can be improved, some of them related with language and some content

Author's response:

We thank the Reviewer for her/his kind interest towards our work. We share his concerns about the disappearance of Enteromates and the difficulty of home chyme reinfusion due to lack of reimbursement.

An overarching comment is that the large number to abbreviations that are difficult to follow so may want to limit them for improvement if the flow of the paper.

Author's answer:

We have removed most of the acronyms created by us: Ld, Lp, Lt, Alb, Cit, CNDA, CFDA except in the tables and figures, explaining their meaning in the legend. We have kept CR, DES and EAF and the acronyms usually used in articles dealing with intestinal failure: IF, IVS PN FE, IFALD, GLP2

Second the impact of the lack of Enteromates pumps needs to be fully explored, personally I have used the technique many times but never without significant resistance from home health provides, this does not seem to be a big factor for the authors and may reflect on this specialized pumps.

Author's answer:

We added the lack of Enteromates pumps L 412-414.

Stopping the commercialization of Enteromates creates a very important difficulty for the diffusion of the technique at the very moment it is recommended. Two teams, to our knowledge, are working on reinfusion devices. Prof. Ian Bisset and Prof. Greg o'Grady in Auckland designed the "Insides System" which is the subject of a BJS article in the April 2020 issue (ref 30). Their device reinfuses the chyme by boluses triggered by the patient or caregivers. Dr. Nzamushe in Lille is working on the validation of the EXCEP system, whose ambition is to be a portable reinfusion automaton (ref 29).

Home care providers do not manage patients who have home RC because they are not familiar with the technique, are not trained to deal with incidents and are not paid by the health insurance. Only patients who demonstrate complete autonomy after receiving therapeutic education have home CR, under our responsibility, with Enteromates that we lend them. We monitor them by telephone, consultations, teleconsultations, in the day hospital if needed. In our area, we are fortunate that some nurses who have worked in our unit and are familiar with RC care, are now working as home care providers. Their work is voluntary and very valuable.

From our point of view, the remuneration of homecare providers by the health insurance should be subject to a care lump sum calculated on the model of those existing in France for parenteral nutrition at home and enteral nutrition at home. A project proposed by us to the health ministry was not retained.

Third please see below:

Line 26 should be 185M not H

Author's answer:

Thanks for seeing it

Line 63 and 99 are L cell enterocytes or enteroneuroendocrine cells

Author's answer:

We changed it

Line 78 the authors have quotations, but I am unsure what is the reference for those quotations.

Line 82 same.

Author's answer:

We reviewed the references and checked

 For table 2 consider changing stomy for stoma.

Author's answer:

We changed it

Line 239 consider using resected instead of amputated.

Author's answer:

We changed it (line 217)

Line 242 what procedure is specifically described as oesogastrectomy,  is a total gastrectomy with distal esophageal resection, a esophagogastrectomy (Ivor Lewis, three field or Tranhiatal type?)

Author's answer:

We changed it (line 220): "Twenty-one (7%) underwent Lewis-Santy oesogastrectomy (3), Finsterer or total gastrectomy and Roux-en-Y reconstruction for cancer (15), gastric bypass (3)."

Line 262 space is missing before the word Prior.

Author's response :

Thank you for seeing it.

Line 272 and line 297 consider changing Stomial for Stomal.

Author's answer:

The spelling has been corrected (lines 244). The paragraph containing stomial line 268 has been amended and the word deleted.

Line 357 to 363 is difficult to read.

Author's answer:

Since the numbering was differenre, we have assumed that this is paragraph that deals with patients xho have had a CR at home. The paragraph has been shortened and the presentation less convoluted (lines 3109-313). The first two lines of the discussion have been modified (lines 317-319).

Line 385 and thereafter consider change the word ingesta for intake although both are accurate.

Author's answer:

We changed it in most cases. We kept the word "ingesta" in three cases where it was specifically for oral feeding.

Reviewer 2 Report

Dear authors

I have to congratulate for the great work done and the research in this area of nutrition. I would like to express a great concern and some comments:

  1. As stated this is an update of previously published data (Picot, Clin Nutr 2017) form your centre. the two manuscript are very similar for contents, and design, with the same tables and figures. In the manuscript is not clear the reason for this update, apparently without any variation but the confirmation of previous results. Please clarify this point exaustively.
  2. Abstract should be completely rewritten in a coincise and clear way. Now it contains too many results and it is too long.
  3. Please reduce the acronyms and abbreviation in the manuscript. They are too many and make the manuscript not easy to read.
  4. In the manuscript the acronym for entero-atmospheric fistula is reported sometimes ad EAF and sometime AEF. please correct.
  5. In the introduction section i think that it should explained more deeply the background, with the clear definitions of DES and EAF (not easy to understand even to surgeons not dedicated to Acute care surgery) and the alreary existing evidences about CR, underlining the research perspectives.
  6. Methods: you declared that the distribution of continuous variable was studied with the KS test; all the variables were normally distributed? if not the should be analyzed with non parametric test as Mann-Withney's U
  7. Which statistical method did you adopt for correlation? 
  8. In table 2: some continue data have #$ (both median and n(%)) please correct. Moreover it would be useful and interesting to have the number of patients analyzed for each variable (since you have high rate of missing data).
  9. Results, line 267: Cit1 is strongly correlate with Cit2. why did you test this correlation? please discuss it
  10. Lines 272-276: are these referred to the figure 3? it's not clear and it is a repetition of methods
  11. Figure 4: the figure is not clear: what are the X and Y axys? what are FE1-2-3-4 andPN 1-2-3-4? please correct!
  12. Line 316: you comment a result about albumin level. please report only data and discuss them in the proper section. 
  13. Figure 3: why did you choose to represent  continuous variables as liver tests as a categorical variable (with no clear cut off)? please change the figure and show results with a boxplot
  14. There is no mention about complications and adverse event. Please add and discuss. The technique is feasible in all patients without any problem? 
  15. What about quality of life?
  16. Line 409: about the FRY study. Is this study ongoing? please provide reference and more details.
  17. In conclusion i think that the paper should be shortened and mada more simple to read.

Author Response

Dear authors

I have to congratulate for the great work done and the research in this area of nutrition. I would like to express a great concern and some comments:

Author's response:

We thank the Reviewer for his kind appreciation of our work. We thank him for his careful reading of our work and have done our best to improve our manuscript in accordance with his concerns

  1. As stated this is an update of previously published data (Picot, Clin Nutr 2017) form your centre. the two manuscript are very similar for contents, and design, with the same tables and figures. In the manuscript is not clear the reason for this update, apparently without any variation but the confirmation of previous results. Please clarify this point exaustively.

Author's response:

We have added a paragraph L98-103.

Allow us for a slightly longer development. It has been known since the 1970s and 1980s that the CR is safe and effective. Our team has demonstrated that this technique, until then reserved for intensive care after digestive surgery, could be useful in follow-up care and we are at the origin of the manufacture of the Enteromates Mobile which have made it possible to do home CR . The recommendations of ASPEN, ESPEN, even for cases of acute IF or IF in ICU, the continued interest of the SFNCM did not change practices.  In France, a few centres are interested in CR but fewer than ten have experience with it. Publications remain very rare, although demonstrative. Caregivers have a priori fear that care is disgusting, and it is disgusting when bad methods are used. French health insurance does not provide for specific remuneration during hospitalization, does not cover the cost of home CR, and refuses to consider it. Home artificial nutrition care providers are waiting for a lump-sum payment to become involved in the home RC.

Parenteral nutrition is effective, its safety has continued to improve, computerized protocols make it easier to prescribe, home care providers and home hospitalization make it possible to be discharged from hospital within a very short period of time, the work of caregivers is well valued, and reimbursement is total, with no outstanding expenses. In many other countries, parenteral nutrition cannot even be offered and mortality is 6 to 10 times higher than what we see. Surgeons do not perform bowel resection if the remaining bowel is likely to be too short and requires PN. In Nepal, it was the drastic reduction in mortality due to chyme reinfusion with a simple syringe that persuaded Ian Bisset to develop his "Insides System" (ref 30).

The current series brings additional results compared to our 2017 article. They confirm the safety and efficacy of CR and correlations are emerging that encourage prospective studies and clinical research. We would like to thank the editors for including our CR experience in the articles in the thematic issue of Nutrients on intestinal failure, and we hope to be more persuasive than we have been to date.

  1. Abstract should be completely rewritten in a coincise and clear way. Now it contains too many results and it is too long.

Author's response:

We have reduced the abstract by 29% by keeping the most demonstrative results.

  1. Please reduce the acronyms and abbreviation in the manuscript. They are too many and make the manuscript not easy to read.

Author's response:

We have removed most of the acronyms created by us: Ld, Lp, Lt, Alb, Cit, CNDA, CFDA except in the tables and figures, explaining their meaning in the legend. We have kept CR, DES and EAF and the acronyms usually used in articles dealing with intestinal failure: IF, IVS PN FE, IFALD, GLP2…

  1. In the manuscript the acronym for entero-atmospheric fistula is reported sometimes ad EAF and sometime AEF. please correct.

Author's response:

Thanks for seeing it

  1. In the introduction section i think that it should explained more deeply the background, with the clear definitions of DES and EAF (not easy to understand even to surgeons not dedicated to Acute care surgery)

Author's response:

We've added a photograph of a double enterostomy in figure 1 and given details in the paragraph L48-51

and the alreary existing evidences about CR, underlining the research perspectives.

Author's response:

At the end of the discussion, we added a paragraph (L 4120-423) indicating the areas of clinical research that CR in humans allows. 

  1.  Methods: you declared that the distribution of continuous variable was studied with the KS test; all the variables were normally distributed? if not the should be analyzed with non parametric test as Mann-Withney's U

Author's response:

 The variables were normally distributed. The paragraph has been modified.

  1. Which statistical method did you adopt for correlation? 

Author's response:

There were Pearson correlations (Lines 193)

  1. In table 2: some continue data have #$ (both median and n(%)) please correct.

Author's response:

Thanks for seeing it

There was an error in the legend of Table 2. We have replaced median cm (25-75% quartiles) by mean M+/- SD. The data in the table are correctly written.

Moreover it would be useful and interesting to have the number of patients analyzed for each variable (since you have high rate of missing data).

Author's response:

The number of patients in each group is in the first line of the table: 269 DES, 37 EAF.

Our error in the legend misled the reviewer. For each line, # indicates the number of measurements, followed in brackets by the percentage that this number represents in each group.

  1. Results, line 267: Cit1 is strongly correlate with Cit2. why did you test this correlation? please discuss it.

Author's response:

Prof. Jacques Cosnes had told me this orally during the discussion of an oral communication that I was presenting to the JFHOD 2011. The discussion of this item was added L 343-350.

  1. Lines 272-276: are these referred to the figure 3? it's not clear and it is a repetition of methods

Author's response:

The layout has been corrected and the legend of the figure has been lightened.

  1. Figure 4: the figure is not clear: what are the X and Y axys? what are FE1-2-3-4 and PN 1-2-3-4? please correct!

Author's response:

The reviewer's right. It's not very readable. We changed the Figure 4. It is presented as a histogram of four classes instead of eight based on daily intravenous supplements volumes.

  1. Line 316: you comment a result about albumin level. please report only data and discuss them in the proper section. 

Author's response:

The reviewer's right. We changed it. The precaution of using albumin level in case of high CRP is very well known and no longer needs to be discussed.  

  1. Figure 3: why did you choose to represent continuous variables as liver tests as a categorical variable (with no clear cut off)? please change the figure and show results with a boxplot.

Author's response:

The figure was presented with boxplots. There was a misnumbering in the legend. It is Figure 5

  1. There is no mention about complications and adverse event. Please add and discuss. The technique is feasible in all patients without any problem? 

Author's response:

A paragraph has been added L 393-407

  1. What about quality of life?

Author's response:

A paragraph has been added L 415-419

  1. Line 409: about the FRY study. Is this study ongoing? please provide reference and more details.

Author's response:

The paragraph has been completed. (L 408-414)

  1. In conclusion I think that the paper should be shortened and mada more simple to read.

Author's response:

The abstract was reduced by 1014 signs and spaces (-29%), material and methods by 2120 signs (-22%), the results by 1987 (-15%). The introduction was lengthened by 367 signs (+8%) and the discussion and conclusion by 2058 signs (+25%) to meet the requests 1, 5, 9, 14, and 15. In total the length was reduced by 2695 signs and spaces (-7%).

Round 2

Reviewer 2 Report

Dear authors,

Thank you very much for your improvements.

It's not clear in the figure legend the x-axis of the figure 4: please add clarifications. what idicates L? 1-2L?

Figure 5: box plots are really better; i would suggest to represent the real values of the variuos parameters (not the number of times the upper value of the normal!)

Author Response

English language and style

 (x) English language and style are fine/minor spell check required ( ) I don't feel qualified to judge about the English language and style

Author's response:

A spell verification found errors and we corrected them.

Comments and Suggestions for Authors

Dear authors

Thank you very much for your improvements.

Author's response:

We thank the reviewer for his/her careful reading of our work. He/she contributed a lot to the improvement of our text and to the deepening of what we wanted to write.

It's not clear in the figure legend the x-axis of the figure 4: please add clarifications. what idicates L? 1-2L?

Author's response:

We added the IVS volume (litres/d) to the right end of the abscissa axis and explain the abbreviation L.

Figure 5: box plots are really better; i would suggest to represent the real values of the variuos parameters (not the number of times the upper value of the normal!)

Author's response:

We have redrawn Figure 5 with the parameter values in UI/L.  We have added a small dotted bar between the paired parameters to indicate the upper value of the normal for each one.